# Determination of piperaquine concentration in human plasma and the correlation of capillary versus venous plasma concentrations

**Norah Mwebaza**[1☯], **Vincent Cheah**[2☯], **Camilla Forsman**[2☯], **Richard Kajubi**[1], **Florence Marzan**[2], **Erika Wallender**[2,3], **Grant Dorsey**[3], **Philip J. Rosenthal**[3], **Francesca Aweeka**[2], **Liusheng Huang**[2]*

**1** Infectious Disease Research Collaboration, Makerere University College of Health Sciences, Kampala, Uganda, **2** Drug Research Unit, Department of Clinical Pharmacy, University of California San Francisco, San Francisco, CA, United States of America, **3** Department of Medicine, University of California San Francisco, San Francisco, CA, United States of America

☯ These authors contributed equally to this work.
* Liusheng.huang@ucsf.edu

**Data Availability Statement:** All relevant data are within the paper and its Supporting Information files.

## Abstract

### Background

A considerable challenge in quantification of the antimalarial piperaquine in plasma is carry-over of analyte signal between assays. Current intensive pharmacokinetic studies often rely on the merging of venous and capillary sampling. Drug levels in capillary plasma may be different from those in venous plasma, Thus, correlation between capillary and venous drug levels needs to be established.

### Methods

Liquid chromatography coupled with tandem mass spectrometry (LC-MS/MS) was used to develop the method. Piperaquine was measured in 205 pairs of capillary and venous plasma samples collected simultaneously at ≥24hr post dose in children, pregnant women and non-pregnant women receiving dihydroartemisinin-piperaquine as malaria chemoprevention. Standard three-dose regimen over three days applied to all participants with three 40mg dihydroartemisinin/320mg PQ tablets per dose for adults and weight-based dose for children. Correlation analysis was performed using the program Stata® SE12.1. Linear regression models were built using concentrations or logarithm transformed concentrations and the final models were selected based on maximal coefficient of determination ($R^2$) and visual check.

### Results

An LC-MS/MS method was developed and validated, utilizing methanol as a protein precipitation agent, a Gemini $C_{18}$ column (50x2.0mm, 5μm) eluted with basic mobile phase solvents (ammonium hydroxide as the additive), and $ESI^+$ as the ion source. This method had

**Funding:** This work is supported by the National Institutes of Health (NIH), grant numbers R01AI117001 (PJR, FA), 4P01HD059454-09 (GD), and P30AI027763 (FA). https://www.nih.gov/grants-funding. The funders had no role in study design, data collection and analysis, decision to publish, or preparation of the manuscript.

**Competing interests:** The authors have declared that no competing interests exist.

a calibration range of 10–1000 ng/mL and carryover was negligible. Correlation analysis revealed a linear relationship: $C_{cap} = 1.04 \times C_{ven} + 4.20$ ($R^2 = 0.832$) without transformation of data, and $\ln C_{cap} = 1.01 \times \ln C_{ven} + 0.0125$, ($R^2 = 0.945$) with natural logarithm transformation. The mean ratio (±SD) of $C_{cap}/C_{ven}$ was 1.13±0.42, and median (IQR) was 1.08 (0.917, 1.33).

## Conclusions

Capillary and venous plasma PQ measures are nearly identical overall, but not readily exchangeable due to large variation. Further correlation study accounting for disposition phases may be necessary.

## Introduction

Piperaquine (PQ) is a long-acting antimalarial drug currently used in combination with dihydroartemisinin for malaria treatment, and also studied for chemoprevention [1, 2]. PQ (Fig 1) is a weak base with pKa values around 8.6 and 6.5 and high lipophilicity (LogP = 6.2) at neutral and alkaline pH [3]. The free base form of PQ is poorly soluble in water, methanol (MeOH), and acetonitrile (MeCN), but very hydrophilic at low pH, and easily soluble in acidified solvents.

Adsorption of PQ to a glass surface occurs if the solution is stored in a glass container, and loss of PQ signal is observed when a metal injection needle or metal tubing is used in an analytical system. Our group found over 50% PQ loss after storage in a glass tube overnight (S1 Table). The extent of adsorption is in the following order: silanized glass > glass > polypropylene plastic tube > Thermo Scientific™ low retention tubes. In plasma samples, over 97% of PQ binds to proteins [4, 5]. These properties present daunting challenges for method development. A considerable challenge is carryover between assays. Carryover could be due to the column (tailing peak), precolumn tubing, or autosampler. Numerous methods have been published for the quantification of PQ in plasma, including high performance liquid chromatography-ultraviolet/visible spectrophotometry (HPLC-UV) [5–7] and liquid chromatography tandem mass spectrometry (LC-MS/MS) [8–12]. Carryover was observed in nearly all LC-MS/MS methods. Here we report an LC-MS/MS method with negligible carryover and its application to venous and capillary samples simultaneously collected for correlation analysis.

Venous blood sampling via catheter remains to be the mainstream method for clinical sample collection. Capillary blood sample collection via finger or heel stick could be used as an

**Fig 1.** Chemical structures of piperaquine (left) and piperaquine-d6 (right).

alternative method, especially in rural areas and pediatric patients [2]. Current intensive pharmacokinetic designs are often relying on the merging of venous and capillary sampling [13, 14]. However, drug levels in capillary plasma may be different from those in venous plasma because of difference in matrix contents, such as oxygen and protein [15, 16]. Correlation between capillary and venous drug levels needs to be established in order to properly interpret and analyze results. A previous study has suggested capillary blood PQ concentrations are higher on average than venous blood PQ in malaria patients [17]. Here we report correlation of venous plasma and capillary plasma PQ concentrations in participants receiving dihydroartemisinin-PQ when enrolled in a malaria chemoprevention trial.

## Methods

### Materials

Piperaquine tetraphosphate tetrahydrate (MW 999.55, purity 99%) was purchased from A.K. Scientific Inc. (Union City, CA, USA). Piperaquine-$d_6$ (PQ-$d_6$, MW 541.55, isotopic purity $\geq$99%) was purchased from AlSAchim, SAS (IllKirch, France). Trichloroacetic acid (TCA, certified ACS reagent) and ammonium formate (NH$_4$FA,certified ACS reagent), trifluoroacetic acid (TFA, Optima™ LC/MS grade), formic acid (FA, Optima™ LC/MS grade) and ammonium hydroxide (NH$_4$OH) (Optima™ LC/MS grade), acetonitrile (MeCN, HPLC grade), methanol (MeOH, HPLC grade), and other common solvents (HPLC grade) were purchased from Fisher Scientific Co. (Fair Lawn, NJ, USA). Blank human plasma (K$_3$EDTA added as anticoagulant) was obtained from Biological Specialty Co (Colmar, PA, USA).

### Quantitation method

Sciex API2000 tandem mass spectrometer was coupled with a Perkin Elmer 200 series micro LC system. The LC column was Gemini C$_{18}$ (50×2.0 mm, 5μm) fitted with a guard column (4×2.0 mm, 5μm) (Phenomenex Inc., Torrance, CA, USA), eluted with 10 mM NH$_4$OH (A) and MeCN (B) in a gradient mode at a flow rate of 0.6 mL/min. ElectroSpray ionization in positive mode (ESI$^+$) was used as the ion source with multiple reaction monitoring (MRM) of $m/z$ 535/288 for PQ and $m/z$ 541/294 for the IS (PQ-$d_6$) for quantitation. PQ stock solution was prepared in MeCN-water (1:9, v/v) containing 0.5% formic acid. Calibration standard samples (10, 25, 50, 100, 250, 500, and 1000 ng/mL) and QC samples (30, 200, and 800 ng/mL) were prepared in blank plasma from two different stock solutions. Plasma samples (25 μL) were mixed with 25 μL 30 ng/mL PQ-$d_6$ in MeCN-water (1:9, v/v) containing 0.5%FA, added 150 μL MeOH, briefly vortex-mixed, and centrifuged at 25,000g for 5min. Transferred ~100μL supernatant to plastic sample vials or 96-well plate in autosampler. Injection volume was 10 μL.

### Validation

The method was validated based on the NIH-sponsored CPQA guidelines [18], which was adopted from the FDA guidelines [19]. A full validation includes precision and accuracy, matrix effect and recovery, dilution integrity and partial volume, stability, concomitant drug interference, and cross validation with a published method. Dilution integrity was evaluated by diluting the extra-high QC sample (3000 ng/mL) by 3-, 5-, and 10-fold with blank plasma. Partial volume was evaluated by mixing 12.5 μL QC samples with 12.5 μL blank plasma. Stability in plasma was evaluated at 4˚C overnight (16hr) in glass vial and after 5 freeze-thaw cycles by comparing the treated samples with untreated samples in plastic microcentrifuge tubes. Long term stability in plasma at -70˚C for 21 months was tested with the published method as

the concentrations of old QC samples were made for the method [3]. To evaluate autosampler stability, the processed low and high QC samples were tested on the same day of processing (as control) and after 7 days in autosampler plastic vials. Stock solution stability was evaluated on UPLC-PDA system by diluting the stock to 10 ug/mL in MeCN-water (1:9) containing 0.5% FA. All measurements were performed in at least triplicates. Matrix effect was evaluated with 7 different lots of human plasma with $K_3$EDTA as the anticoagulant. Set 1 samples were prepared by spiking both PQ and IS in 75% MeOH with final concentrations of 3.75, 25, and 100 ng/mL for PQ and 3.75 ng/mL for IS, corresponding to the final concentrations of PQ and IS from plasma samples after protein precipitation. Set 2 samples were spiked at the same concentration as Set 1 in extracted solutions from 7 lots of blank plasma, and Set 3 samples were prepared by spiking PQ in 7 lots of blank plasma with a final PQ concentration of 30, 200, and 800 ng/mL and then processing the plasma samples as described above.

### Clinical sample analysis

The method applied to capillary versus venous plasma PQ correlation study, which is part of a pharmacokinetic study within the larger trials for malaria chemoprevention in pregnant women and children (ClinicalTrials.gov number, NCT02163447) [13, 14, 20, 21]. The study was conducted in Tororo, Uganda from December 2014 to May 2017. Eligible participants were pregnant women with ultrasound-estimated gestational age of 12–20 weeks and their children. Complete entry criteria were summarized previously [13, 14, 22]. The studies were approved by the Uganda National Council of Science and Technology and institutional review boards of Makerere University and the University of California, San Francisco. Written informed consent was obtained from adult study participants, and, for children, from their parents or guardians. The reported method was used to analyze 150 pairs of plasma samples from capillary and venous blood simultaneously collected 24 hr post last dose. We also modified a previously published method in our group to a lower calibration range of 0.5–50 ng/mL [3], and performed the required partial validation for this modification (S1 File). The modified method was used for 65 pairs of plasma samples from capillary and venous blood simultaneously collected from 7 to 84 days post 1st dose. Standard three-dose regimen over three days applied to all participants with three 40mg dihydroartemisinin/320mg PQ tablets per dose for adults [13] and weight-based dose for children [14].

### Correlation analysis

Using STATA SE12.1, the relationship between capillary and venous plasma PQ concentrations was modelled using a linear relationship with estimated intercept and slope. The linear least squares regression models were built using concentrations or logarithm transformed concentrations and the final models were selected based on maximal coefficient of determination ($R^2$) and visual check.

## Results

### 1. Method development

The optimized MS parameters were shown in Table 1. Most published methods for analysis of PQ have utilized acidic mobile phase solvents. Alternatively, basic mobile phase solvents can be used. Lindegardh's group published a method using 2.5 mM ammonium bicarbonate (pH10.0)-acetonitrile (15:85, v/v) as the mobile phase solvents [8]. Based on this method, we developed a new method on an API2000 system. A tailing peak was observed when we tried to use the same solvents [23], so we decided to use ammonium hydroxide in the mobile phase.

**Table 1. Optimized MS parameters.**

| Source parameters | TEM, ˚C | IS, v | CAD, psi | CUR, psi | Gas1, psi | Gas2, psi | |
|---|---|---|---|---|---|---|---|
| | 500 | 2000 | 11 | 30 | 40 | 70 | |
| Compound parameters | DP, v | FP, v | EP, v | CE, v | CEP, v | CXP, v | Dwell time, ms |
| 535/288 (PQ) | 76 | 360 | 12 | 49 | 20 | 10 | 250 |
| 541/294 (PQ-$d_6$, I.S.) | 76 | 360 | 12 | 49 | 20 | 10 | 250 |

TEM, source temperature; IS, ionspray voltage; CUR, curtain gas; Gas1, nebulizer gas; Gas2, auxiliary gas; CAD, collision-activated dissociation; DP, declustering potential; FP, focusing potential; EP, entrance potential; CE, collision energy; CEP, collision cell entrance potential; CXP, collision cell exit potential.

Higher $NH_4OH$ concentration led to better peak shape, but lower signal intensity. Optimal peaks were observed using 10 mM $NH_4OH$ as mobile phase A, and 100% acetonitrile as mobile phase B (Fig 2). The final LC gradient program consists of 55% solvent B (0 min), from 55 to 100% B (0–1.5 min), 100% B (1.5–2.0 min), 100%-55% B (2.0–2.1 min), and 55% B (2.1–2.5 min). Surprisingly, no carryover peaks were observed with this method, though occasionally a small carryover peak was found in some runs, probably due to pH variation of mobile phase A. Ammonia may evaporate and carbon dioxide in air may dissolve in mobile phase A to change the pH over time. Therefore, we recommend that mobile phase A be prepared freshly every day before use. Disappearance of carryover peak may also possibly because the API2000 system is less sensitive.

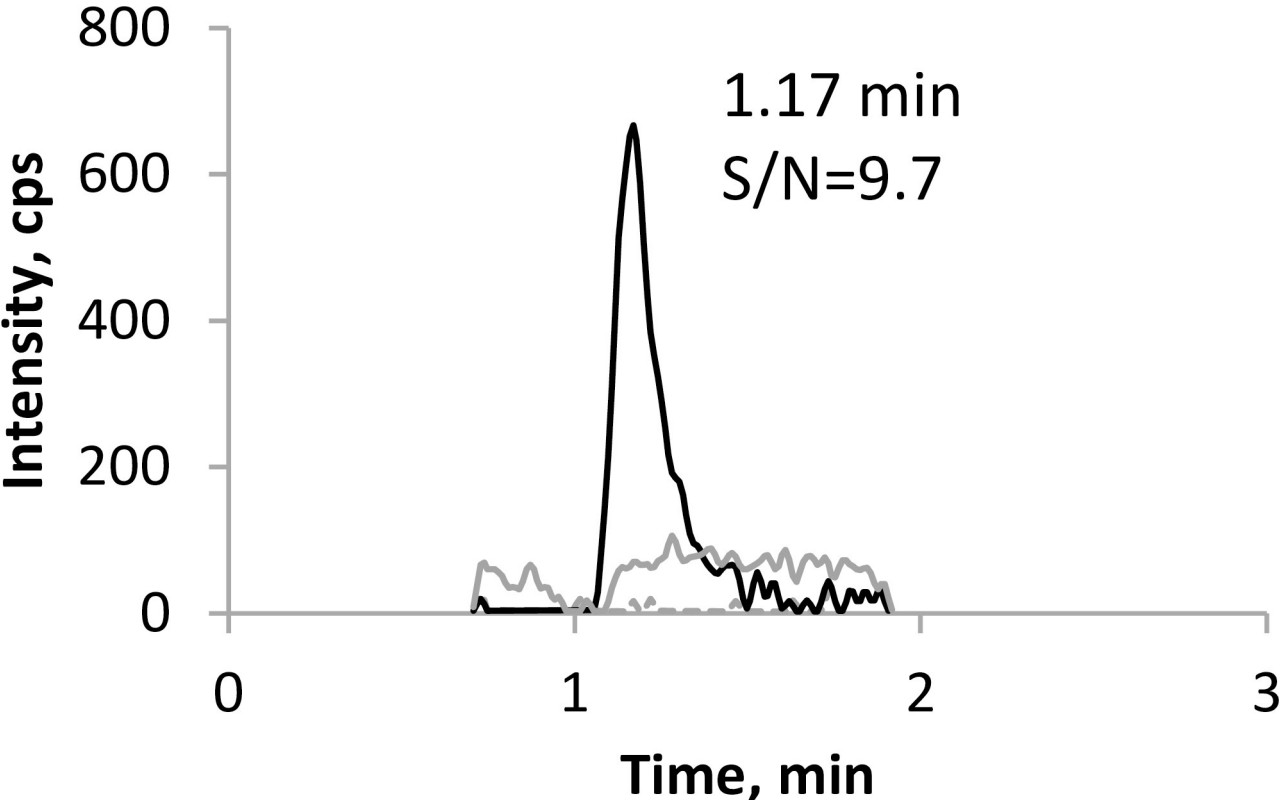

**Fig 2.** Chromatograms of PQ representing blank plasma (dashed gray line), the LLOQ (solid black line) and blank plasma after ULOQ, indicating carryover (solid gray line).

**Table 2. Intra- and inter-day precision and accuracy.**

| Nominal, ng/mL | Intra-day (n = 6) | | | | Inter-day (n = 18) | | | |
|---|---|---|---|---|---|---|---|---|
| | **10.0** | **30.0** | **200** | **800** | **10.0** | **30.0** | **200** | **800** |
| mean, ng/mL | 9.97–10.2 | 30.1–32.7 | 202–217 | 828–853 | 10.1 | 31.5 | 211 | 842 |
| RSD,% | 4.8–8.3 | 2.4–4.0 | 2.2–3.6 | 1.2–3.3 | 6.2 | 4.6 | 4.2 | 2.7 |
| %dev | (-0.32)-2.3 | 0.4–9.1 | 0.8–8.5 | 3.5–6.6 | 0.6 | 5.1 | 5.5 | 5.3 |

## 2. Method validation

The lower limit of quantitation (LLOQ) for this assay was set at 10 ng/mL, with a S/N ratio = 9.7. The calibration range was 10–1000 ng/mL. The calibration curve was fitted with least square linear regression weighted by 1/x. The correlation coefficient (r) was typically > 0.9990. Representative chromatograms of PQ from blank plasma extract, LLOQ, and blank plasma extract injected after the upper limit of quantitation (ULOQ) are shown in Fig 2.

The intra-day precision (n = 6) over 3 days ranged from 1.2 to 4.0% at the three concentrations (30, 200, and 800 ng/mL), and inter-day precisions ranged from 2.7 to 4.6%, all of them within 15% deviation from mean values. The intra- and inter-day accuracy ranged from 0.4 to 9.1%and 5.1 to 5.3%, respectively, all within 15% deviation from the nominal values. At the LLOQ 10 ng/mL level, the precision and accuracy met the criteria of <20% (Table 2).

Absolute matrix effect was evaluated with mean peak areas from Set 1 (clean solution) and Set 2 (post-extraction spiked solution). A value of 100% indicated no matrix effect. If the value was >100%, ion enhancement was observed, and if <100%, ion suppression was observed. At low, medium, and high concentrations, the matrix effect for PQ was 107, 86.6, and 91.8%, respectively (Table 3). The IS normalized matrix effect (PQ/IS) was close to 100%. These results indicated that the matrix effect was not significant and compensated well by the deuterated IS.

The recovery of PQ at 30, 200, and 800 ng/mL was 68.4, 85.5, and 87.3% respectively. The mean recovery for the I.S. was 83.0%.

To determine dilution integrity, we diluted an extra high QC (3000 ng/mL) by 3-, 5-, and 10-fold. The accuracy (% deviation) for dilution samples ranged from -4.4 to 2.2 (S2 Table). We also tested partial volumes by taking half volume samples mixed with an equal volume of blank plasma. The % deviation for all partial volume samples ranged from 1.3 to 9.9 (S2 Table). These results confirm that plasma samples can be diluted by up to 10-fold.

Previously, we reported that PQ was stable in plasma at room temperature (21–24˚C) for 9 days, at -70˚C for 8 months, and after 3 freeze-thaw cycles [3]. In this report, we further tested and found that PQ is stable in plasma after 5 freeze-thaw cycles and at -70˚C for at least 21 months. There was no difference in results for plasma samples prepared in glass versus plastic vials, likely due to PQ bound to plasma proteins. The processed samples were stable in auto-sampler plastic vials for at least 7 days, and stock solution was stable at -70˚C for at least 3

**Table 3. Matrix effect and recovery (n = 7).**

| Conc (ng/ml) | PQ Peak Area, x10⁴ | | | IS Peak Area, x10⁴ | | | Matrix Effect | | | Recovery | |
|---|---|---|---|---|---|---|---|---|---|---|---|
| | **Set 1** | **Set 2** | **Set 3** | **Set 1** | **Set 2** | **Set 3** | **PQ** | **IS** | **PQ/IS** | **PQ** | **IS** |
| Low (30) | 4.13±1.38 | 4.40±0.62 | 3.01±0.43 | 4.30±0.15 | 4.26±0.36 | 3.28±0.33 | 107 | 99.1 | 108 | 68.4 | 77.0 |
| Med (200) | 29.0±2.5 | 25.1±2.7 | 21.4±1.2 | 4.41±0.36 | 3.88±0.42 | 3.23±0.23 | 86.6 | 88.0 | 98.4 | 85.5 | 83.3 |
| High (800) | 110±3 | 101±6 | 878±7 | 4.15±0.12 | 3.66±0.35 | 3,24±0.29 | 91.8 | 88.2 | 104 | 87.3 | 88.6 |

Set 1, MeOH-Water (3:1, v/v) solution; Set 2, post extraction spiked solution; Set 3, pre-extraction spiked solution.

**Table 4. Stability of PQ. Data represent mean (± SD).**

| Conditions | Untreated[***] | treated | % remained | n |
|---|---|---|---|---|
| Plasma, glass container, 4˚C, 16 hr | | | | |
| 30 ng/mL | 29.8±2.3 | 30.2±1.1 | 101 | 3 |
| 800 ng/mL | 796±4 | 832±14 | 105 | 3 |
| Autosampler plastic vials, 21–25˚C, 7 days | | | | |
| 30 ng/mL | 32.8±1.7 | 29.4±1.1 | 89.6 | 4 |
| 800 ng/mL | 865±27 | 857±19 | 99.1 | 4 |
| Five freeze-thaw cycles | | | | |
| 30 ng/mL | 31.7±1.3 | 30.9±1.2 | 97.5 | 3 |
| 800 ng/mL | 844±12 | 843±12 | 99.9 | 3 |
| Plasma, -70˚C, 21 months[*] | | | | |
| 3 ng/mL | 2.76±0.15 | 2.78±0.04 | 101 | 3 |
| 200 ng/mL | 200±6 | 209±5 | 104 | 3 |
| Stock, -70˚C, 40 months[**] | | | | |
| | 8170±605 | 8091±338 | 99 | 3 |

[*] Measured with our previously published method

[**] measured with UPLC-PDA with peek tubing injection needle.

[***] untreated samples were prepared fresh in plastic vials.

years (Table 4). Noticeably, evaluation of stock solution stability with the API2000 yielded large variation from repeated injections; this is most likely due to the metal injection needle used for the API 2000 system. When switched to an Acquity UPLC-PDA system with a poly-etheretherketone (PEEK) sample needle, reproducible results were obtained.

To test potential concomitant drug interference, lumefantrine, artemether, dihydroartemisinin, nevirapine, efavirenz, zidovudine, lamivudine, stavudine (D4T), lopinavir, nelfinavir, indinavir, saquinavir, and amprenavir were spiked into the medium QC samples. The differences of the spiked samples compared to the control medium QC were within 5%, confirming no interference from potential concomitant drugs (S3 Table).

To further validate the API2000 method, we re-analyzed 116 samples that were analyzed with the previously published method [3]. The concentration of those samples ranged from 9.06 to 553 ng/mL. Three samples were measured as below LLOQ. Among the 113 quantifiable samples, 100 samples (88.5%) were within 20% of the reference value (S4 Table).

## 3. Carryover

Having 6 nitrogen atoms with different pKa values, PQ can be found in multiple forms in solution. Due to this property, peak tailing and carryover peaks were often observed during the analysis of PQ. The majority of published LC-MS/MS methods utilized acidic mobile phase solvents [3, 9–12], while two methods used basic mobile phase solvents [8, 23] (Table 5). Carryover peaks were observed in nearly all published LC-MS/MS methods. Most methods maintained the carryover peaks within 20% LLOQ. In the method published by Hodel et al, the carryover peak of PQ was >100% LLOQ [10], partially due to the wide calibration range. Recently, our group published a method on an API 5000 LC-MS/MS system [3] in which carryover from the column was removed, but carryover from the autosampler/injection was still present at ~0.08% of ULOQ (250 ng/mL). Accordingly, the LLOQ was raised to 1.5 ng/mL. In the modified method with a calibration range of 0.5-50ng/mL, the residual carryover peak was maintained within 20% LLOQ (S3 File). Our group also developed a method using a basic

**Table 5. Published methods for PQ quantification.**

| Reference | Instrument | Column | Mobile phase | Sample preparation | Calibration range | Carryover | Retention factor k |
|---|---|---|---|---|---|---|---|
| Hodel et al, 2009 | TSQ Quantum (ESI+) | Atlantis dC18 (50x2.1mm) | 20mM $NH_4FA$ 0.5% FA; MeCN 0.5%FA | PPT: 200uL plasma +700uL MeCN | 2–4000 ng/mL | yes (>100% LLOQ) | 9.7 |
| Singhal et al, 2007 | API4000 Q-trap (ESI+) | Chromolith SpeedROD RP-18e (50x4.6mm) | $NH_4AC$-MeOH-FA-$NH_3$ | PPT: 50uL plasma +300uL MeOH | 1–250 ng/mL | yes (not specified) | 0.6 |
| Lindegardh, et al, 2008 | API5000 (ESI+) | Gemini C18 (50x2.0mm) | $NH_4HCO_3$-MeCN | SPE: 50 uL plasma | 1.5–500 ng/mL | <15%LLOQ | 2.4 |
| Lee et al, 2011 | API2000 (ESI+) | Zorbax C18 (50x2.1mm, 5um) | $NH_4HCO_3$-MeCN | Dilution: 50uL PBS samples +IS | 20-1000ng/mL | yes (18–125%LLOQ) 4 | |
| Kjellin, et al, 2014 | API5000 (APCI+) | Pursuit PFP (50x2.0mm, 3um)) | $NH_4FA$-TFA-MeCN | PPT: 25uL plasma +100uL MeOH-TCA | 1.5-250ng/mL | <20%LLOQ | 4 |
| Liu et al, 2017 | API5500 Q-trap (ESI+) | Venusil XBP-C18 (50x2.1mm, 5um) | 2mMNH4AC 0.15%FA 0.05% TFA-MeCN | PPT: 40uL plasma +HCl+160 uL MeCN | 2–400 ng/mL | ≤LLOQ | 3 |
| Aziz et al, 2017 | API4000 (ESI+) | Express C18 (30x2.1mm, 5um) | 0.1%FA in Water-MeCN | PPT: 100uL plasma + 300uL 1%FA in water | 3.9–2508 nM | Yes (not specified) | 29 |
| This report | API5000 (APCI+) | Pursuit PFP (50x2.0mm, 3um) | $NH_4FA$-TFA-MeCN | PPT: 25uL plasma +100uL MeOH-TCA | 0.5-50ng/mL | <20%LLOQ | 4 |
| This report | API2000 (ESI+) | Gemini C18 (50x2.0mm,5um) | 10mM$NH_4OH$-MeCN | PPT: 25uL plasma +25ul IS+150uL MeOH | 10–1000 ng/mL | no | 6 |

mobile phase [10], based on a method published by Lindegardh et al [2]. The peak tailing and carryover were significant (S3 File). In the current report, we changed the basic additive from 2.5 mM $NH_4HCO_3$ to 10 mM $NH_4OH$, and changed the column from Zorbax Eclipse $C_{18}$ (50x2.1mm, 5μm) to Gemini $C_{18}$ (50x2.0, 5μm). With these changes, the carryover peak was reduced markedly. Although carryover peak was observed in some runs, in most cases no carryover peaks were observed with this method (S4 File). It might possibly be due to the less sensitivity of the instrument and/or the narrow calibration range spanning only 100-fold when compared to other methods. It is also likely due to the basic mobile phase condition. We speculate that the ionic PQ molecule causes the carryover, which could diminish at basic condition where PQ is uncharged. Considering PQ pka = 8.6, PQ will be uncharged if mobile phase pH >10.6, and analytical columns applicable at pH 11 or higher are now available with the advancement of column technology.

## 4. Analysis of clinical samples

A total of 215 pairs of plasma samples simultaneously collected from capillary and venous blood were analyzed. Five pairs of samples were below LLOQ (0.5ng/mL) for both capillary and venous PQ, 4 capillary samples were below LLOQ with measurable venous PQ, and 1 venous sample was below LLOQ with measurable capillary PQ, yielding a total of 205 pairs of quantifiable data points for correlation analysis.

## 5. Correlation of PQ in capillary versus venous plasma

A total of 205 pairs of data above LLOQ were obtained from samples collected from 24 hr to 82 days post last dose in children, pregnant women and non-pregnant adults. Simple linear regression yields an equation: $C_{cap} = 1.04 \times C_{ven} + 4.20$, $R^2 = 0.832$ (Table 6). The 95% confidence intervals (CI) of the intercept included zero (-0.878, 9.27), and the p-value was 0.105, suggesting the difference from zero was not statistically significant. The slope was 1.04 with 95% CI (0.978, 1.11). The results suggest PQ concentrations in capillary and venous plasma

**Table 6. Correlation of capillary and venous PQ plasma concentrations.**

|  | Total (n = 205) | 24 hr post last dose (n = 150) | ≥ 7 days post dose (n = 55) |
|---|---|---|---|
| $C_{ven}$, ng/mL | 50.4 (0.504, 251) | 70.8 (9.67, 251) | 3.41 (0.504, 20.4) |
| $C_{cap}$, ng/mL | 56.3 (0.584, 292) | 73.5 (16.7, 292) | 2.84 (0.584, 22.6) |
| P value | <0.0001 | <0.0001 | 0.38 |
| Correlation equition: $C_{cap} = a \times C_{ven} + b$ | | | |
| a | 1.04 (0.978, 1.11) | 0.984 (0.886, 1.08) | 0.965 (0.828, 1.10) |
| b | 4.20 (-0.878, 9.27) | 10.6 (1.68, 19.5) | -0.182 (-1.02, 0.661) |
| $R^2$ | 0.832 | 0.729 | 0.789 |

Concentrations represent medians (range) and correlation parameters represent means (95%CI).

appear to be correlated in a simple linear relationship. However, large variation led to a scattered correlation plot (Fig 3), a significant portion of samples could not be explained by the linear equation, making extrapolation of PQ concentrations from capillary to venous plasma complicate. With natural log-transformed data, improved correlation was obtained with $R^2$ values of 0.945 and the equation $lnC_{cap} = 1.01 \times lnC_{ven} + 0.013$. In general, slightly higher PQ concentrations were found in capillary samples, with a median value of 56.3 ng/mL versus 50.4

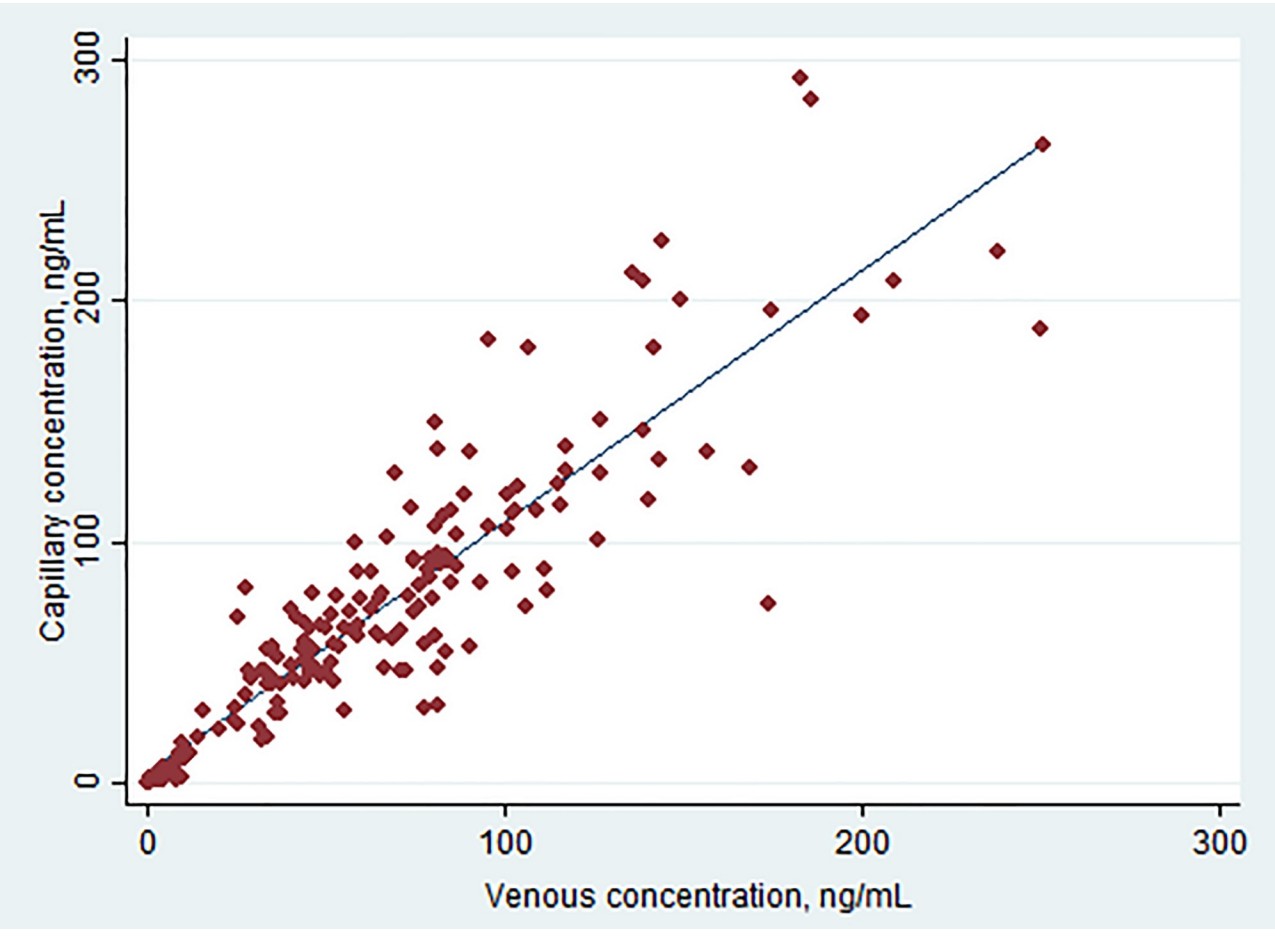

**Fig 3. Linear regression of capillary versus venous plasma PQ.**

ng/mL in venous samples (p<0.0001). The mean ratio (±SD) of $C_{cap}/C_{ven}$ was 1.13±0.42, and median with interquartile range (IQR) was 1.08 (0.917, 1.33). However, 78 of 205 capillary PQ values (38%) were lower than corresponding venous PQ values.

Considering the impact of pharmacokinetic stages, we performed sub-analysis with data from 24 hr post last dose and ≥ 7 days post dose. At 24 hr post last dose, we collected 150 pairs of data, 57 from pregnant women, 63 from children under 2 years old, and 30 from non-pregnant women. The correlation equation at this time interval in each population was published previously with natural log-transformed data, which yielded better correlation than non-transformed data [13, 14]. When all data at 24 hr post last dose were combined (n = 150), simple linear regression yielded a coefficient of determination ($R^2$) of 0.729 (Table 6). We observed a slightly higher capillary PQ concentration on average, with the median value of 73.5 ng/mL versus 70.8 ng/mL for venous PQ (p<0.0001). The mean ratio (±SD) of $C_{cap}/C_{ven}$ was 1.17 ±0.37, and median (IQR) was 1.12 (0.952, 1.34). The data suggest that the capillary concentration is slightly higher at the distribution phase. However, 48 of the 150 capillary PQ concentrations (32%) were lower than venous PQ concentrations. Natural log-transformed data did not improve the correlation. At ≥ 7 days post dose, we collected 55 pairs of data from children, yielding a linear regression equation $C_{cap} = 0.965 \times C_{ven} - 0.182$ and $R^2 = 0.789$. Log-transformed data did not improve the correlation. The median PQ concentrations for venous and capillary plasma were 3.41ng/mL and 2.84ng/mL, respectively, and the difference was not statistically significant (p = 0.38). The mean ratio (±SD) of $C_{cap}/C_{ven}$ was 1.03±0.52, and median (IQR) ratio was 0.974 (0.793, 1.21). Over 50% of capillary PQ concentrations were lower than venous PQ concentrations, suggesting that at the terminal elimination phase venous PQ tends to be higher than capillary PQ. A study in Burkina Faso reported day 7 PQ concentrations in capillary plasma were higher than those in venous plasma with median (range) concentration at 67 (49–84) in capillary versus 41 (27–59) in venous plasma (n = 186, p<0.001) [24]. Our result is different, likely because 54 of the 55 pairs of samples were collected beyond 20 days after dose, which are better representatives of the terminal elimination phase.

A study in Thailand estimated correlation of dihydroartemisinin-piperaquine administration on malaria patients aged >2 years [17]. Venous blood PQ were nearly always higher than venous plasma PQ with a median (90% range) ratio of 2.15 (0.91, 5.26), suggesting PQ concentrated in red blood cells; while the difference between capillary blood and venous blood PQ concentrations was smaller with median (90% range) ratio of 1.66 (0.92, 3.03). After day 3 when parasitaemia had cleared, a simple relationship was found: the venous blood PQ concentration = (capillary blood PQ concentration)$^{0.9}$. However, similar to what we observed here, there were also large variations in that study, leading to the conclusion that measurements of venous and capillary PQ concentration are not readily interchangeable. In contrast, another antimalarial, lumefantrine, showed good linear correlation at a 1:1 ratio between capillary and venous samples, with coefficient of determination $R^2 > 0.95$ [21]. This is probably due to the complex pharmacokinetic profile of PQ, such as multiple peak concentrations, slow distribution of PQ requiring longer time to reach equilibrium, large distribution volume (up to 874L/kg), and long elimination half-life (up to 28 days) [25]. It is likely not feasible to find a linear correlation during early pharmacokinetic phases, as equilibrium between different compartments is not reached and varied among individuals. Correlation may be better accessed during the steady state and the elimination phase., e.g day 14 or 28.

## Conclusions

We reported a method for quantitation of PQ in plasma with a calibration range of 10–1000 ng/mL. Carryover was negligible in the method.

The concentrations of PQ in capillary and venous plasma were correlated in a linear relationship. However, due to large variations, exchange of PQ concentrations between capillary and venous plasma will compromise precision of the results. Correlation study accounting for disposition phases may be necessary.

## Supporting information

**S1 Table. Adsorption of PQ on container surface.**
(XLSX)

**S2 Table. Dilution and partial volume precision and accuracy for PQ.**
(XLSX)

**S3 Table. Interference of potential concomitant drugs.**
(XLSX)

**S4 Table. Cross validation using the previous API5000-based method as reference.**
(XLSX)

**S1 File. Partial validation of the modified PQ assay.**
(PDF)

**S2 File.**
(DTA)

**S3 File.**
(PDF)

**S4 File.**
(PDF)

## Acknowledgments

We wish to thank the volunteers participating in this study, the study coordinators at the participating sites, and supporting staff at Drug Research Unit at UCSF.

## Author Contributions

**Conceptualization:** Francesca Aweeka.

**Data curation:** Norah Mwebaza, Camilla Forsman, Erika Wallender.

**Formal analysis:** Camilla Forsman, Liusheng Huang.

**Funding acquisition:** Grant Dorsey, Philip J. Rosenthal, Francesca Aweeka.

**Investigation:** Norah Mwebaza, Vincent Cheah, Richard Kajubi, Florence Marzan, Erika Wallender, Liusheng Huang.

**Methodology:** Liusheng Huang.

**Project administration:** Norah Mwebaza.

**Resources:** Grant Dorsey.

**Supervision:** Francesca Aweeka.

**Validation:** Vincent Cheah, Liusheng Huang.

**Writing – original draft:** Camilla Forsman, Liusheng Huang.

**Writing – review & editing:** Norah Mwebaza, Richard Kajubi, Erika Wallender, Grant Dorsey, Philip J. Rosenthal, Francesca Aweeka.

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
