## [Decision Letter · Decision Letter 0]

10 Mar 2020

PONE-D-20-05889

Determination of piperaquine concentration in human plasma and the correlation of capillary versus venous plasma concentrations

PLOS ONE

Dear Dr Huang,

Thank you for submitting your manuscript to PLOS ONE. After careful consideration, we feel that it has some merit but does not meet PLOS ONE’s publication criteria as it currently stands.  In particular, the expert reviewer has major concerns about the fact that the authors cannot justify the absence of carry-over in their method by showing additional results (see reviewer's requests in the comments to the author),  Consequently the statements and conclusions in the manuscript are not sufficiently supported by the results. If this can not be appropriately addressed in a revision we will move to rejection of the manuscript. We now invite you to submit a revised version of the manuscript that addresses the points raised during the review process.

We would appreciate receiving your revised manuscript by Apr 24 2020 11:59PM. To enhance the reproducibility of your results, we recommend that if applicable you deposit your laboratory protocols in protocols.io, where a protocol can be assigned its own identifier (DOI) such that it can be cited independently in the future. For instructions see: http://journals.plos.org/plosone/s/submission-guidelines#loc-laboratory-protocols

We look forward to receiving your revised manuscript.

Kind regards,

Henk D. F. H. Schallig, Ph.D

Academic Editor

PLOS ONE

Journal Requirements:

2. In your Methods section, please provide additional information about the participant recruitment method. Please ensure you have provided sufficient details to replicate the analyses such as: a) the recruitment date range (month and year), b) a description of any inclusion/exclusion criteria that were applied to participant recruitment, and c) a description of how participants were recruited.

Reviewers' comments:

Reviewer's Responses to Questions

**Comments to the Author**

1. Is the manuscript technically sound, and do the data support the conclusions?

Reviewer #1: No

2. Has the statistical analysis been performed appropriately and rigorously? 

Reviewer #1: I Don't Know

3. Have the authors made all data underlying the findings in their manuscript fully available?

Reviewer #1: No

4. Is the manuscript presented in an intelligible fashion and written in standard English?

Reviewer #1: No

5. Review Comments to the Author

Reviewer #1: The authors describe a novel chromatography method to separate and eventually quantify piperaquine in human plasma, plus an additional comparison of venous plasma concentrations versus capillary plasma concentrations. Regarding the detection method, at least 5 other piperquine methods have been described in the literature, and also the comparison of venous versus capillary concentrations has extensively been described, I believe one of the first papers was in 2010 from Lindegardh's group.

The authors propose that the method described here is the first method that does not exhibit carry-over, a characteristic which is present for piperaquine given its high adsorbance to a variety of surfaces. However, the authors appear to describe two methods in the current paper, a novel one with a new chromatography and a range of 10-1000 ng/mL, and an older one, previously published in 2014, with a range of 0.5-50 ng/mL. This is actually very unclear in the manuscript, LLOQ's are constantly being mixed up and also the methodology only appears to describe the method of 10-1000 ng/mL, while little details are given regarding the method of 0.5-50 ng/mL. I have a few major queries and comments:

(1) The main point of the new method (10-1000 ng/mL) appears to be the avoidance of carry-over, apparently achieved according to the authors, by the new chromatography. Carry-over can be easily avoided by injecting blank samples after a high concentrations. The only benefit of avoiding carry-over would be that the order of measuring samples does not need to be accounted for, which would be an added benefit. However, the another second method was used by the authors, using a more sensitive machine (API5000 instead of API2000) to measure the low concentrations, which was actually previously published already (but then with an LLOQ of 1.5 ng/mL). Th application of this second method for concentrations in the lowest range still requires the analyst to know which samples contain which concentrations, to know which samples should be measured at which machine. This means that all lower concentrations are not even measured with the method, in the same assay, as any of the higher concentrations, and thus the added value of having no carry-over in the method with the 10-1000 ng/mL is, in my opinion, absent.

(2) Perhaps more importantly, I have the impression that the absence of noticing a carry-over with the new method (10-1000 ng/mL) is an artefact of the methodology used and the high limit of quantification of this method. Actually, as the authors showed in their previous publication on piperquine LCMSMS analysis from 2014 in Bioanalysis, the carry-over after an ULOQ using the method that they used for quantifying the concentration range of 0.5-50 ng/mL, was only 0.2 ng/mL. This means that the LLOQ of the new method (10 ng/mL) is about 50 times the signal of the carry-over. Given that the API2000 is a much less sensitive machine and the high LLOQ with this method, I simply assume that a carry-over of maximally 0.2 ng/mL is simply not visible with this unsensitive methodology. I would be happy to see my statements corrected by any additional evidence provided by the authors, but based on what currently is provided the statement that carry-over is absent with the new chromatography is not supported by the results.

(3) More details are needed regarding the origin and quality of chemicals. Moreover, I suggest the authors to adhere to the international bioanalytical validation requirements as issued by FDA and/or EMA.

(4) The ULOQ seems to be high for the encountered concentrations in the clinical samples (maximally 250-300 ng/mL), why and how was chosen for this calibration range?

(5) Differences between venous and capillary plasma concentrations have previously been published, however these are not well discussed in the current discussion.

(6) The authors suggest a linear relationship between venous and capillary plasma concentrations. The fit of this relationship appears to be far from optimal and extrapolation of venous concentrations from capillary concentrations appears difficult, the manuscript would benefit if the authors could acknowledge this limitation and discuss its impact.

(7) Regarding this linear relationship between venous and capillary concentrations: the authors state in the discussion: "This is probably due to the complex PK profile of PQ, such as multiple peak concentrations and slow distribution of PQ requiring longer time to reach equilibrium." If this would be the case a linear relationship over the whole concentration range seems physiologically not correct, the manuscript would benefit if the authors could consider and discuss this.

(8) Ethics: the registration numbers relating to approval by any of the ethical/institutional review boards should preferably be included in the manuscript for full ethical transparency.

6. PLOS authors have the option to publish the peer review history of their article (what does this mean?). If published, this will include your full peer review and any attached files.

Reviewer #1: No

---

## [Author Response · Author response to Decision Letter 0]

24 Apr 2020

Editor's comments:

Response: It was written referring to the templates

2. In your Methods section, please provide additional information about the participant recruitment method. Please ensure you have provided sufficient details to replicate the analyses such as: a) the recruitment date range (month and year), b) a description of any inclusion/exclusion criteria that were applied to participant recruitment, and c) a description of how participants were recruited.

Response: The information for clinical trial was published previously and can be found in the reference. The following sentences are added now: “(ClinicalTrials.gov number,NCT02163447) [13, 14, 19, 20]. The study was conducted in Tororo, Uganda from December 2014 to May 2017. Eligible participants were pregnant women with ultrasound-estimated gestational age of 12-20 weeks and their children. Complete entry criteria were summarized previously[13, 14, 21].”

3. We note that you have indicated that data from this study are available upon request. PLOS only allows data to be available upon request if there are legal or ethical restrictions on sharing data publicly. For more information on unacceptable data access restrictions, please see http://journals.plos.org/plosone/s/data-availability#loc-unacceptable-data-access-restrictions

Response: the concentration data for the correlation analysis is now provided in the supplemental file (S2 file.dta).

Reviewer #1: The authors describe a novel chromatography method to separate and eventually quantify piperaquine in human plasma, plus an additional comparison of venous plasma concentrations versus capillary plasma concentrations. Regarding the detection method, at least 5 other piperquine methods have been described in the literature, and also the comparison of venous versus capillary concentrations has extensively been described, I believe one of the first papers was in 2010 from Lindegardh's group.

The authors propose that the method described here is the first method that does not exhibit carry-over, a characteristic which is present for piperaquine given its high adsorbance to a variety of surfaces. However, the authors appear to describe two methods in the current paper, a novel one with a new chromatography and a range of 10-1000 ng/mL, and an older one, previously published in 2014, with a range of 0.5-50 ng/mL. This is actually very unclear in the manuscript, LLOQ's are constantly being mixed up and also the methodology only appears to describe the method of 10-1000 ng/mL, while little details are given regarding the method of 0.5-50 ng/mL. I have a few major queries and comments: 

Response: Thank you for your careful review. The focus of this manuscript is on the new method. The modification of the older method with a lower LLOQ (0.5 ng/mL) was only briefly described but validation data can be found in supporting information (S1 file). Method details can be found in the older method published in Bioanalysis. 

(1) The main point of the new method (10-1000 ng/mL) appears to be the avoidance of carry-over, apparently achieved according to the authors, by the new chromatography. Carry-over can be easily avoided by injecting blank samples after a high concentrations. The only benefit of avoiding carry-over would be that the order of measuring samples does not need to be accounted for, which would be an added benefit. However, the another second method was used by the authors, using a more sensitive machine (API5000 instead of API2000) to measure the low concentrations, which was actually previously published already (but then with an LLOQ of 1.5 ng/mL). The application of this second method for concentrations in the lowest range still requires the analyst to know which samples contain which concentrations, to know which samples should be measured at which machine. This means that all lower concentrations are not even measured with the method, in the same assay, as any of the higher concentrations, and thus the added value of having no carry-over in the method with the 10-1000 ng/mL is, in my opinion, absent.

Response: I agree that impact of carryover can be avoided by injecting blank samples after a higher concentration. However, this is inconvenient for unknown samples, especially if we submit an overnight run, reinjections are required on the following day. The new method (10-1000ng/mL) was developed to support intensive PK studies with expected Cmax up to 1000 ng/mL. We analyzed ~2000 intensive PK samples with this new method. Only samples at elimination tail end were analyzed with the modified older method. There are 22% (455 out of 2075) samples above 250ng/mL, the ULOQ of our older published method. By avoiding carryover and dilution, we minimize reanalysis to save samples and efforts. This is a significant benefit, especially for plasma samples collected from pediatric participants, the sample volume is very limited.

(2) Perhaps more importantly, I have the impression that the absence of noticing a carry-over with the new method (10-1000 ng/mL) is an artefact of the methodology used and the high limit of quantification of this method. Actually, as the authors showed in their previous publication on piperquine LCMSMS analysis from 2014 in Bioanalysis, the carry-over after an ULOQ using the method that they used for quantifying the concentration range of 0.5-50 ng/mL, was only 0.2 ng/mL. This means that the LLOQ of the new method (10 ng/mL) is about 50 times the signal of the carry-over. Given that the API2000 is a much less sensitive machine and the high LLOQ with this method, I simply assume that a carry-over of maximally 0.2 ng/mL is simply not visible with this unsensitive methodology. I would be happy to see my statements corrected by any additional evidence provided by the authors, but based on what currently is provided the statement that carry-over is absent with the new chromatography is not supported by the results.

Response: Thanks for the thoughtful comments. The API2000 is indeed 20-100 fold less sensitive than API5000 depending on types of compounds. But please keep in mind, carryover amount is associated with the ULOQ. The carryover of 0.2ng/mL is based on the ULOQ of 250ng/mL and lower carryover was observed with a ULOQ of 50 ng/mL (Supporting S3 file). I project a carryover of ~0.8ng/mL following injection of a 1000ng/mL samples with the older method. Indeed, we observed significant carryover in an earlier in vitro study with API 2000 system using a calibration range of 20-1000ng/mL (Supporting S3 file). We also observed carryover sometimes in the new method (See supporting materials S4 file, page 1), but in most cases, there is no carryover (Supporting S4 file). Whereas, we did notice the sensitivity of the API2000 varied in different days. 

(3) More details are needed regarding the origin and quality of chemicals. Moreover, I suggest the authors to adhere to the international bioanalytical validation requirements as issued by FDA and/or EMA.

Response: Sentences were reworded to clarify the quality of chemicals (line 88-91). We validated methods based on CPQA guidelines, which is based on the FDA guidelines, but with more restrictions. For example, Stock solution used for QC samples needs to a separately prepared solution from the stock solution used for calibrators. In FDA guidelines, this is not required as long as the stock solution was verified to be accurate. 

(4) The ULOQ seems to be high for the encountered concentrations in the clinical samples (maximally 250-300 ng/mL), why and how was chosen for this calibration range?

Response: please see response to question #1.

(5) Differences between venous and capillary plasma concentrations have previously been published, however these are not well discussed in the current discussion.

Response: discussion and citation are added now. The following sentences were added in line 309 to 314: “A study in Burkina Faso reported day 7 PQ concentrations in capillary plasma were higher than those in venous plasma with median (range) concentration at 67 (49-84) in capillary versus 41 (27-59) in venous plasma (n=186, p<0.001). Our result is different, likely because 54 of the 55 pairs of samples were collected beyond 20 days after dose, which are better representatives of the terminal elimination phase.”

One additional reference was found and cited: Zongo I, Some FA, Somda SA, Parikh S, Rouamba N, Rosenthal PJ, et al. Efficacy and day 7 plasma piperaquine concentrations in African children treated for uncomplicated malaria with dihydroartemisinin-piperaquine. PLoS One. 2014;9(8):e103200. doi: 10.1371/journal.pone.0103200. PubMed PMID: 25133389; PubMed Central PMCID: PMCPMC4136730.

(6) The authors suggest a linear relationship between venous and capillary plasma concentrations. The fit of this relationship appears to be far from optimal and extrapolation of venous concentrations from capillary concentrations appears difficult, the manuscript would benefit if the authors could acknowledge this limitation and discuss its impact.

Response: thanks for the suggestion. We write additional discussion now in Line 273-276 as follows: “ The results suggest PQ concentration in capillary and venous plasma is likely at 1:1 ratio. However, Large interindividual variation led to a scattered correlation plot (Figure 3), a significant portion of samples are not at 1:1 ratio, making extrapolation of PQ concentration from capillary to venous plasma difficult.” 

(7) Regarding this linear relationship between venous and capillary concentrations: the authors state in the discussion: "This is probably due to the complex PK profile of PQ, such as multiple peak concentrations and slow distribution of PQ requiring longer time to reach equilibrium." If this would be the case a linear relationship over the whole concentration range seems physiologically not correct, the manuscript would benefit if the authors could consider and discuss this.

Response: Thanks for the comments, additional discussion is added to reflect our opinion that correlation may be better at the late elimination phase. The following sentences were added in line 329-333:” It is likely not feasible to find a good correlation during early pharmacokinetic phases, as equilibrium between different compartments is not reached and varied among individuals. Better correlation may be obtained if correlation was performed at single time points in the elimination phase., e.g day 14 or 28.”

(8) Ethics: the registration numbers relating to approval by any of the ethical/institutional review boards should preferably be included in the manuscript for full ethical transparency.

Response: The clinical trials.gov registration number is provided (NCT02163447).

---

## [Decision Letter · Decision Letter 1]

30 Apr 2020

PONE-D-20-05889R1

Determination of piperaquine concentration in human plasma and the correlation of capillary versus venous plasma concentrations

PLOS ONE

Dear Dr Huang,

Thank you for submitting your manuscript to PLOS ONE. After careful consideration, we feel that it has merit but does not fully meet PLOS ONE’s publication criteria as it currently stands. Therefore, we invite you to submit a revised version of the manuscript that addresses the points raised during the review process.

The expert reviewer has assessed your revised manuscript. Unfortunately, you did not address or meet all the required revisions requested by the reviewer. You have now the oportunity to correct this further. We want to receive a clear response to the issues raised by the reviewer and you must address all points raised.

We would appreciate receiving your revised manuscript by Jun 14 2020 11:59PM. To enhance the reproducibility of your results, we recommend that if applicable you deposit your laboratory protocols in protocols.io, where a protocol can be assigned its own identifier (DOI) such that it can be cited independently in the future. For instructions see: http://journals.plos.org/plosone/s/submission-guidelines#loc-laboratory-protocols

We look forward to receiving your revised manuscript.

Kind regards,

Henk D. F. H. Schallig, Ph.D

Academic Editor

PLOS ONE

Reviewers' comments:

Reviewer's Responses to Questions

**Comments to the Author**

1. If the authors have adequately addressed your comments raised in a previous round of review and you feel that this manuscript is now acceptable for publication, you may indicate that here to bypass the “Comments to the Author” section, enter your conflict of interest statement in the “Confidential to Editor” section, and submit your "Accept" recommendation.

Reviewer #1: (No Response)

2. Is the manuscript technically sound, and do the data support the conclusions?

Reviewer #1: Yes

3. Has the statistical analysis been performed appropriately and rigorously? 

Reviewer #1: Yes

4. Have the authors made all data underlying the findings in their manuscript fully available?

Reviewer #1: Yes

5. Is the manuscript presented in an intelligible fashion and written in standard English?

Reviewer #1: Yes

6. Review Comments to the Author

Reviewer #1: I thank the authors for their revisions. I have a few more comments, the numbering below refers to my original comments and the authors' replies to these points:

(2) I believe the authors acknowledge my observation regarding the lower sensitivity of the equipment which is probably the reason for not observing the carry-over, instead of the adjusted chromatography. The authors suggest that carry-over would amount to 0.8 ng/mL for a 1000 ng/mL sample, this would indeed mean that carry-over remains undetected with the method with LLOQ of 10 ng/mL, but nevertheless might still be present. This needs to be clearly acknowledged in the manuscript as the authors still suggest now throughout the manuscript that carry-over is avoided due to the new chromatography method, a statement which I do not see supported by any data.

(6) This statement regarding the ratio between plasma and venous samples would benefit from mentioning some quantification about the variability, e.g. an easy way to illustrate the level of variability would be mentioning the range/IQR of the quantified ratio's for each paired sample set. Please also remove the word 'interindividual' as the variability is not due to variability between patients, but rather between sample time points (at least this is what the authors suggest).

(7) I assume the authors intended here 'linear correlation' and not just 'correlation'. Additionally, correlation might be better assessed during steady-state of the pharmacokinetics.

7. PLOS authors have the option to publish the peer review history of their article (what does this mean?). If published, this will include your full peer review and any attached files.

Reviewer #1: No

---

## [Author Response · Author response to Decision Letter 1]

13 May 2020

Reviewer #1: 

(2) I believe the authors acknowledge my observation regarding the lower sensitivity of the equipment which is probably the reason for not observing the carry-over, instead of the adjusted chromatography. The authors suggest that carry-over would amount to 0.8 ng/mL for a 1000 ng/mL sample, this would indeed mean that carry-over remains undetected with the method with LLOQ of 10 ng/mL, but nevertheless might still be present. This needs to be clearly acknowledged in the manuscript as the authors still suggest now throughout the manuscript that carry-over is avoided due to the new chromatography method, a statement which I do not see supported by any data.

Response: We want to keep the explanation open to multiple posibilities. We observed carryover peak during early validation, but in later experiments and routine sample analysis the carryover peak diminished, we don’t know the exact reasons for this improvement. I acknowledge the possibility that reviewer brought up and now add the following sentences:

Line 176-177: Disappearance of carryover peak may also possibly because the API2000 system is less sensitive.

Line 260-260: It might possibly be due to the less sensitivity of the instrument and/or the narrow calibration range spanning only 100-fold when compared to other methods. It is also likely due to the basic mobile phase. We speculate that the ionic PQ molecule causes the carryover, which could diminish at basic condition where PQ is uncharged. Considering PQ pka=8.6, PQ will be uncharged if mobile phase pH >10.6, and analytical columns applicable at pH 11 or higher are now available with the advancement of column technology.

(6) This statement regarding the ratio between plasma and venous samples would benefit from mentioning some quantification about the variability, e.g. an easy way to illustrate the level of variability would be mentioning the range/IQR of the quantified ratio's for each paired sample set. Please also remove the word 'interindividual' as the variability is not due to variability between patients, but rather between sample time points (at least this is what the authors suggest).

 Response: The mean and median ration with IQR are described later in the paragraph. I reworded the sentences slightly as follows: 

Line 278-287“ The results suggest PQ concentrations in capillary and venous plasma appear to be correlated in a simple linear relationship. However, large variation led to a scattered correlation plot (Figure 3), a significant portion of samples could not be explained by the linear equation, making extrapolation of PQ concentrations from capillary to venous plasma complicate……. The mean ratio (±SD) of Ccap/Cven was 1.13±0.42, and median with interquartile range (IQR) was 1.08 (0.917, 1.33). However, 78 of 205 capillary PQ values (38%) were lower than corresponding venous PQ values.”

Line 354-356: due to large variations, exchange of PQ concentrations between capillary and venous plasma will compromise precision of the results.

(7) I assume the authors intended here 'linear correlation' and not just 'correlation'. Additionally, correlation might be better assessed during steady-state of the pharmacokinetics.

 Response: Thanks for your constructive suggestion. The sentences are updated as follows.

It is likely not feasible to find a linear correlation during early pharmacokinetic phases,….. . Correlation may be better accessed during the steady state and the elimination phase…

---

## [Decision Letter · Decision Letter 2]

15 May 2020

Determination of piperaquine concentration in human plasma and the correlation of capillary versus venous plasma concentrations

PONE-D-20-05889R2

Dear Dr. Huang,

We are pleased to inform you that your manuscript has been judged scientifically suitable for publication and will be formally accepted for publication once it complies with all outstanding technical requirements.

With kind regards,

Henk D. F. H. Schallig, Ph.D

Academic Editor

PLOS ONE

Additional Editor Comments (optional):

Reviewers' comments:

Reviewer's Responses to Questions

**Comments to the Author**

1. If the authors have adequately addressed your comments raised in a previous round of review and you feel that this manuscript is now acceptable for publication, you may indicate that here to bypass the “Comments to the Author” section, enter your conflict of interest statement in the “Confidential to Editor” section, and submit your "Accept" recommendation.

Reviewer #1: All comments have been addressed

2. Is the manuscript technically sound, and do the data support the conclusions?

Reviewer #1: Yes

3. Has the statistical analysis been performed appropriately and rigorously? 

Reviewer #1: Yes

4. Have the authors made all data underlying the findings in their manuscript fully available?

Reviewer #1: Yes

5. Is the manuscript presented in an intelligible fashion and written in standard English?

Reviewer #1: Yes

6. Review Comments to the Author

Reviewer #1: (No Response)

7. PLOS authors have the option to publish the peer review history of their article (what does this mean?). If published, this will include your full peer review and any attached files.

Reviewer #1: No

---

## [Editor Report · Acceptance letter]

21 May 2020

PONE-D-20-05889R2 

Determination of piperaquine concentration in human plasma and the correlation of capillary versus venous plasma concentrations 

Dear Dr. Huang:

I am pleased to inform you that your manuscript has been deemed suitable for publication in PLOS ONE. Congratulations! Your manuscript is now with our production department. 

With kind regards,

on behalf of

Dr. Henk D. F. H. Schallig 

Academic Editor

PLOS ONE